# Digital Health Technologies for Optimising Treatment and Rehabilitation Following Surgery: Device-Based Measurement of Sling Posture and Adherence

**DOI:** 10.3390/s25010166

**Published:** 2024-12-31

**Authors:** Joss Langford, Ahmed Barakat, Engy Daghash, Harvinder Singh, Alex V. Rowlands

**Affiliations:** 1ActivInsights Ltd., 6 Nene Road, Bicton Industrial Park, Kimbolton, Huntingdon PE28 0LF, UK; joss@activinsights.com; 2Faculty of Health and Life Sciences, University of Exeter, Stocker Rd, Exeter EX4 4PY, UK; 3Trauma & Orthopaedics Department, University Hospitals of Leicester NHS Trust, Leicester LE5 4PW, UK; ahmedharoonbarakat@gmail.com (A.B.); engy.daghash@uhl-tr.nhs.uk (E.D.); hps9@leicester.ac.uk (H.S.); 4School of Healthcare, University of Leicester, Leicester LE1 7RH, UK; 5Assessment of Movement Behaviours (AMBer), Leicester Lifestyle and Health Research Group, Diabetes Research Centre, University of Leicester, Leicester LE5 4PW, UK; 6National Institute for Health Research, Leicester Biomedical Research Centre, Leicester LE5 4PW, UK; 7Alliance for Research in Exercise, Nutrition and Activity (ARENA), UniSA Allied Health and Human Performance, Division of Health Sciences, University of South Australia, Adelaide 5000, Australia

**Keywords:** shoulder, sling, adherence, accelerometer, surgery, wrist, arm, posture

## Abstract

Background: Following shoulder surgery, controlled and protected mobilisation for an appropriate duration is crucial for appropriate recovery. However, methods for objective assessment of sling wear and use in everyday living are currently lacking. In this pilot study, we aim to determine if a sling-embedded triaxial accelerometer and/or wrist-worn sensor can be used to quantify arm posture during sling wear and adherence to sling wear. Methods: Four participants were asked to wear a GENEActiv triaxial accelerometer on their non-dominant wrist for four hours in an office environment, and, for two of those hours, they also wore a sling in which an additional GENEActiv accelerometer was secured. During sling wear, they were asked to move their arm in the sling through a series of pre-specified arm postures. Results: We found that upper arm angle and posture type during sling wear can be predicted from a sling sensor alone (R^2^ = 0.79, *p* < 0.001 and Cohen’s kappa = 0.886, respectively). The addition of a wrist-worn sensor did not improve performance. The optimisation of an existing non-wear algorithm accurately detected adherence (99.3%). Conclusions: the remote monitoring of sling adherence and the quantification of immobilisation is practical and effective with digital health technology.

## 1. Introduction

Post-operative rehabilitation following shoulder surgery, such as rotator cuff repairs or shoulder arthroplasties, usually involves wearing a sling to immobilise the shoulder [1,2]. The adherence to sling wearing has two components: sling wear duration and posture of the arm when wearing the sling.

The primary aim of sling wear is the immobilisation of the arm for an appropriate duration. Premature mobilisation can lead to failure of repair while protracted immobilisation can lead to stiffness and a frozen shoulder, both detrimental to function [3,4,5]. No consensus exists on the optimal duration of sling wear, as current evidence is conflicting, with recommended durations ranging from no immobilisation at all to six weeks of sling wear [6]. Further, perfect patient adherence to both the prescribed duration of sling wear and immobilisation posture is often assumed. Some studies have used patient questionnaires to assess adherence to sling wear [7]; however, self-report tends to lead to over-estimation of adherence relative to objective assessment [8,9].

In addition to whether the sling is worn, how much the arm is immobilised during sling wear is also likely to impact rehabilitation. If the shoulder is immobilised completely during the early post-shoulder-surgery-rehabilitation phase in the sling while the tendons are healing, there is risk of stiffness and prolonging recovery [4]. However, if the shoulder could be mobilised only within a safe zone [10], the recovery may be more rapid. There is a trend towards early mobilisation after shoulder surgery [11] where early joint motion within a safe zone could prevent significant stress on the surgical repair or injury and lead to a better range of motion and greater patient satisfaction. However, the characterisation of the safe zone for mobilisation would benefit from further evidence. An objective assessment of the shoulder range of motion could inform the refinement of the recommended safe zone, be helpful for patient confidence, and potentially lead to better functional outcomes.

We recently developed and validated a digital health technology (DHT, FDA, 2023) approach for objectively determining adherence to sling wear duration [2]. Our method for determining sling wear utilised acceleration and temperature measured by a sensor integrated in the sling. The assessment of both acceleration and temperature mitigated the weaknesses of either approach alone, e.g., the time lag for the increase or drop in temperature following sling donning or doffing, respectively, or the continued movement following sling removal due to a sling being carried in a bag [2].

Further, the development of sling-embedded accelerometery methods could enable the assessment of movement and arm posture during sling wear. This is possible as, when accelerometers are moving, they provide a very good measure of activity intensity [12]. However, when an accelerometer is static or moving slowly, their detection of the Earth’s gravitational field allows the determination of the orientation of the device and, hence, the posture of the body part or object to which they are attached [12,13,14].

Therefore, DHTs based on a sensor integrated in the sling and/or a sensor worn on the wrist could provide information that enables determination of the arm posture when the sling is being worn. Combined with the sling wear algorithm, this would enable the determination of both components of adherence: sling wear duration and arm posture, or degree of immobilisation, when wearing the sling. This application of DHT could directly inform the development of evidence-based optimal treatment plans and rehabilitation based on sling wear and a safe zone for mobilisation following shoulder surgery [15,16].

A three-component framework intended to provide a foundational evaluation framework for DHTs has been proposed [17], including three steps: verification, analytical validation, and clinical validation (V3). The engineering verification of our selected sensor system has already been well-described [18]. Here, we used a pilot study to address the analytical validity of potential new digital measures by comparing processed sensor-level data to clinician observation (our criterion measure), and then we explored the application of this approach in a free-living office environment.

### 1.1. Aim

The aim of this study was to determine if a triaxial accelerometer embedded in the sling and/or worn on the wrist can be used to determine arm posture during sling wear and adherence to sling wear.

### 1.2. Hypotheses

A sling sensor will give a direct measure of upper arm elevation and may enable the estimation of the type of arm posture during sling wear (neutral/forward flexion/side abduction/external rotation).The addition of a wrist-worn sensor will not improve the assessment of upper arm elevation but may improve the assessment of the type of arm posture.A sling or wrist-worn sensor individually, or in combination, can describe arm posture during sling wear with a low burden in real-world settings.Sling wear can be predicted from the sling sensor using a previously described algorithm.Arm postures consistent with sling wear can be predicted through an assessment of position and dynamics from the wrist-worn sensor alone.

## 2. Materials and Methods

Four volunteers (recruited through the Leicester University Medical Research Society (LUMRS) platform) provided written informed consent prior to data collection. Volunteers were approached and recruited through the Leicester University Medical Research Society. Participants with no ongoing shoulder problems nor previous history of problems with their shoulders or surgery were eligible. This study received ethical approval from the National Research Ethics Committee (REC) Integrated Research Application System (IRAS, reference: 315132). The volunteers understood the aim of the study and were fully briefed beforehand.

Participants were asked to wear a triaxial accelerometer (GENEActiv, ActivInsights, Cambridgeshire, UK) (Figure 1a) on their dominant wrist for 4 h in an office environment, during which they were asked to go about their normal office activities (e.g., desk work, office tasks, and sporadic walking). For 2 out of the 4 h, the participants wore a universal shoulder immobilisation sling (BeneCare Poly Arm Sling) on the same arm (dominant) in which an additional GENEActiv accelerometer was secured (Figure 1b). All participants were right-handed and wore the wrist accelerometer and sling on the right arm.

The sensors were placed in an inside pocket of the sling positioned at the elbow in line with the humerus bone as in Barakat et al. [2]. This ensured it accurately picked up shoulder motion (movement of the humerus bone in relation to shoulder blade/scapula).

During the 2 h period, participants were asked to move their arm in the sling through a series of nine pre-specified arm postures in a randomised order. They were asked to hold each posture for one minute, return to neutral for 1 min, before moving to the next posture. Participants were trained in each of the nine arm postures (Figure 2) prior to the study starting, and the length of the sling band was adjusted so that the elbow joint would be at 90 degrees when resting with gravity in the sling. The nine postures were as follows:Neutral: 10° upper arm elevation.Forward flexion: 30°, 60°, and 90° upper arm elevation.Side abduction: 30°, 60°, and 90° upper arm elevation.External rotation with the arm adducted (0° degrees upper arm elevation): 0° and 30° rotation.

Participants recorded sling wear, donning, doffing times, and the exact start and end time for each arm posture to accurately label the accelerometer data. Although self-reported adherence to sling wear can be unreliable [15,16], this study asked healthy volunteers to follow a 4 h non-wear and arm position schedule. This required them to immediately log the exact times of sling donning and doffing and of each arm posture in a logbook, avoiding any recall bias.

The GENEActiv weighs 16 g and measures 43 × 40 × 13 mm. It contains a MEMS triaxial accelerometer with a dynamic range of +/− 8 *g* and a 12-bit (3.9 m*g*) resolution and a near-body linear active thermistor with a range of 0–60 °C and a resolution of 0.25 °C. At the maximum measurement frequency of 100 Hz used in this study, the data are stored locally on the device for up to seven (version 1.1) to fourteen days (version 1.2) on a single charge. The orientation of the x, y, and z axes when the GENEActiv is wrist-worn and when embedded in the sling is shown in Figure 1.

The accelerometers were initialised in GENEActiv PC Software (version 3.3) to collect data at 100 Hz. The data were downloaded and then processed in R (version 4.4.1, 14 June 2024) where 1 s epoch summaries were created. The epoch length was selected to maximise the time resolution of the analysis while providing smoothing for extraneous movement signals. The summaries included the mean and standard deviation of acceleration in each axis, mean absolute gravity-subtracted acceleration, and mean temperature [2]. For data from the wrist, the mean rotation and lower arm elevation were also calculated [14]. The data were labelled using the participant logged times for sling wear/non-wear and arm posture.

Thus, while there were only 4 participants, the analyses were based on ~240 sets of datapoints per arm position (~60 per participant) and ~56,000 sets of datapoints across the data collection period (~14,000 per participant).

### 2.1. Data Analysis

The protocol generated three nested datasets for each participant (Figure 3), each with sling and wrist sensor data.

(a)Pre-specified arm posture while wearing sling;(b)Sling wear in free-living ‘office environment’;(c)Sling wear and non-wear in free-living ‘office environment’.

All statistical processing was completed in R (version 4.4.1, 14 June 2024) using base R linear models (e.g., model1 = lm(angle~mean_y, data = dataset_a)).

#### 2.1.1. Hypothesis 1: A Sling Sensor Will Give a Direct Measure of Upper Arm Angle and May Enable Estimation of the Type of Arm Posture During Sling Wear (Neutral/Forward Flexion/Side Abduction/External Rotation)

Dataset (a), sling sensor only.

Model 1: Linear regression, prediction of upper arm angle from mean acceleration in y axis (vertical).Model 2: Model 1 plus mean acceleration in x and z axes.Model 3a: Principal component analysis (PCA) of mean acceleration in x, y, and z axes followed by a linear regression, prediction of upper arm angle from principal components PC1 and PC2.Model 3b: PCA of mean acceleration in x, y, and z axes followed by a linear regression, prediction of upper arm angle from just PC1.Model 4: PCA of mean acceleration in x, y, and z axes followed by cluster analysis, prediction of arm posture during sling wear.

#### 2.1.2. Hypothesis 2: A Wrist-Worn Sensor, Alone or in Combination with a Sling Sensor, Will Not Improve the Assessment of Upper Arm Angle, but May Improve Assessment of Type of Arm Posture During Sling Wear

Dataset (a), sling and wrist sensor.

Model 5a: Linear regression, prediction of upper arm angle from wrist elevation and rotation.Model 5b: Linear regression, prediction of upper arm angle from sling sensor PC 1, wrist elevation and rotation.Model 6: PCA of sling mean acceleration in x, y, and z axes, wrist elevation and wrist rotation, prediction of arm posture during sling wear.

#### 2.1.3. Hypothesis 3: A Sling or Wrist-Worn Sensor Individually, or in Combination, Can Describe Arm Posture During Sling Wear with Low Burden in Real-World Settings

Dataset (b) excluding dataset (a), sling and/or wrist sensor.

Model 7: Model 4 applied to sling wear during ‘free-living’ office environment to determine sling posture from sling sensor.Model 8: Prediction of arm posture during sling wear (including upper arm angle) from wrist elevation and rotation.Model 9: Model 6 applied to sling wear during ‘free-living’ office environment.

#### 2.1.4. Hypothesis 4: Sling Wear Can Be Predicted from the Sling Sensor Using a Previously Described Algorithm

Dataset (c), sling sensor only.

Implementation and enhancement of algorithm from Baraket et al. as a state machine [2].

#### 2.1.5. Hypothesis 5: Arm Postures Consistent with Sling Wear Can Be Predicted Through an Assessment of Position and Dynamics from the Wrist-Worn Sensor Alone

Dataset (c), wrist sensor only.

Comparison of distributions of parameters derived from wrist acceleration.

## 3. Results

All participants completed the programme with no reports of discomfort relating to the placement of the sensor within the sling.

### 3.1. Hypothesis 1

Dataset (a), sling sensor only.

The acceleration measured in the y axis of the sling sensor was significantly associated with upper arm angle in the specified sling postures, R^2^ = 0.75 (*p* < 0.001, Model 1, Figure 4). The addition of the x and z axes of acceleration increased the R^2^ to 0.83 (*p* < 0.001, Model 2).

The first and second components of the PCA explained over 90% of the relative importance, with the loadings primarily from the y and x axes (Table 1). PC1 explained 79.2% of the variance in upper arm angle with R^2^ = 0.79 (*p* < 0.001) and no improvement from adding PC2 (Model 3a), thus only PC1 (Model 3b) was selected for use in further analyses for the prediction of upper arm angle.

The clustering of components 1 and 2 revealed distinct patterning by type of movement (neutral, forward flexion, side abduction, and external rotation) (Model 4, Figure 5). Both PC1 and PC2 were retained for analyses of arm posture during sling wear, with increasing PC1 indicating increasing upper arm elevation. The Dunn Index is the ratio of the minimum distance between clusters and the maximum distance within clusters with higher scores representing better clustering; for Model 4, we found a value of 0.619.

A simple, visual classification schema between clusters was defined as shown in the dashed lines in Figure 5. The level of agreement between the clinician observations and classification schema as assessed by Cohen’s kappa was almost perfect at 0.886.

### 3.2. Hypothesis 2

Dataset (a), sling and wrist sensor.

Wrist elevation (Figure 6a) and wrist rotation (Figure 6b) explained ~30% less variance in upper arm angle than the acceleration from the sling sensor, R^2^ = 0.52 (Model 5a).

Adding PC1 of the sling sensor data increased the variance explained to 0.81 (Model 5b), giving no additional benefit compared to using the sling sensor alone (Model 2, R^2^ = 0.83).

Further, when wrist sensor data were added to the model of sling sensor data alone, there was no improvement in the Dunn Index (0.585) for arm posture type during sling wear (Model 6). Therefore, the original Model 4 was retained for the analyses of sling posture, and Model 9 was not run.

### 3.3. Hypothesis 3

Dataset (b) excluding dataset (a), sling sensor.

Clustering PC1 and PC2 from the sling sensor data revealed distinct patterning during sling wear while in a free-living office environment (Model 7, Figure 7). Applying the sling posture classification schema from Model 4 suggests the majority of the time was spent in the forward flexion (67.1%) and external rotation postures (18.5%) (Table 2).

The mean standard deviation was used to assess the average movement in each cluster as it was highly correlated with the mean acceleration intensity (0.88), and the standard deviation is more sensitive in low movement scenarios (i.e., sling wear). The mean standard deviation was lowest in the forward flexion posture (below the 13 m*g* typically used to assess non-movement [12]) and highest in the neutral and side abduction postures.

Examining the arm posture during sling wear by participant suggested high variability. This can be observed in participant-specific frequency plots of upper arm angles predicted by the sling sensor (Figure 8). The variability between participants was evident with participant 4 spending a high proportion of the time with the upper arm elevated: upper arm angle > 45° (see Figure 2 for the illustration of upper arm angles).

Dataset (b) excluding dataset (a), wrist sensor.

In contrast to the sling sensor data (Model 7), wrist rotation and elevation from the wrist sensor did not show any distinct clusters that would obviously reflect arm posture during sling wear (Model 8, Figure 9).

### 3.4. Hypothesis 4

Dataset (c), sling sensor only.

The algorithm developed to detect sling non-wear by Baraket et al. [2] was adapted for implementation in R as a state machine (Figure 10). An increase or decrease in temperature accompanied by a change from low to high acceleration standard deviation (or vice versa) triggers the change of wear state. Following initial testing, the original algorithm was modified to remove the requirement for lack of movement to mark the beginning of a wear period. This was to avoid movement in removing the sling leading to misclassification.

The predictive performance of both algorithms is shown in Table 3. The optimisation improved the sensitivity markedly to give a final F-score of 0.99 and accuracy of 99.3%.

An illustration of the optimised algorithm in action is shown in Figure 11. Green shows the movement rule (mean standard deviation > 0.013 g (13 m*g*)), blue shows the temperature dropping (<−0.2°), suggesting the start of a non-wear period, while red shows the temperature increasing (>0.1°), suggesting the end of non-wearing and the start of wear. The rules combined give the dark trace at the top of the plot, indicating wear and non-wear.

### 3.5. Hypothesis 5: Arm Postures Consistent with Sling Wear Can Be Predicted Through an Assessment of Position and Dynamics from the Wrist-Worn Sensor Alone

Dataset (c), wrist sensor only.

Differences on the distributions of wrist acceleration, wrist rotation, and wrist elevation can be clearly identified in sling wear (left panel, Figure 12) and non-wear (right panel, Figure 12). During sling wear, the mean acceleration was lower (top panel, Figure 12), the extremes of wrist elevation were reduced (middle panel, Figure 12), and the range of wrist rotations was curtailed (bottom panel, Figure 12). When sling wear and non-wear were presented together with predicted arm postures during sling wear (Figure 13), the constraint to movement was immediately apparent. However, there was considerable overlap between the clusters of sling wear and non-wear.

### 3.6. Summary of Results

Overall, we found that a sling sensor provided a direct measure of upper arm elevation and enabled the estimation of the type of arm posture during sling wear (hypothesis 1 was true). The addition of a wrist-worn sensor did not improve the assessment of the type of arm posture (hypothesis 2 was false).

The sling described arm postures during sling wear with a low burden in real-world settings (hypothesis 3 was true), and sling wear could be predicted from the sensor (hypothesis 4 was true). We had insufficient data due to the design of the study to assess whether arm postures consistent with sling wear could be predicted from a wrist-worn sensor alone (hypothesis 5 was inconclusive).

## 4. Discussion

We found that a sling sensor provided a good estimation of upper arm angle and arm posture during sling wear. The selected model (a principal component analysis of the three axes of acceleration where PC1 corresponded to upper arm angle and the clustering of PC1 and PC2 predicted type of arm posture during sling wear) proved the simplest, well-performing model with good generalisability (minimal risk of model overfit).

The predictive model allowed the assessment of time spent in different arm postures during sling wear during free-living. The deployment of this DHT with patients in real-world settings during treatment and rehabilitation will enable clinical validation of this DHT (V3, FDA, 2023; Goldsack et al. [17]). This validation would seek to assess the health outcomes associated with the two components of sling wear adherence: duration and arm posture. A further step would be to assess the adherence to and further evidence for the safe zones of movement.

The safe zone positions [10] describe arm postures during sling wear that prevent significant stress on surgical repair or injury. Standard safe zones are generally up to 90° of forward flexion and 10° of external rotation. However, safe zone mobilisation needs further development and any sensor-based DHT that provides feedback on the shoulder range of motion zone, which will be helpful for patient confidence and lead to better functional outcomes. Personalised and dynamic safe zones in such a system could be specified from the quality of repair at the time of surgery by the surgeon or by a therapist during rehabilitation. In the future, real-time feedback to patients may also be supportive of recovery [19].

The methodology of this study could be extended to consider the range of motion (ROM) assessment as a key component of functional outcome measures. ROM is typically assessed either visually or with a goniometer, with the latter offering greater accuracy. A sensor-based system attached directly to the upper arm would overcome the challenges of conventional goniometric measurements: operator-dependent, time-consuming, and not facilitating continuous monitoring. Continuous monitoring is particularly valuable in evaluating a patient’s functional capacity, both pre- and post-operatively. In the pre-operative phase, pre-operative physiotherapy or what is known as “prehabilitation” has recently mounted increasing interest. Prehabilitation postulates that enhancing prior to surgery can lead to improved perioperative outcomes, as the prehabilitated patient may retain higher functional ability and recover more quickly than those who did not undergo prehabilitation [20].

Traditional methods of measuring adherence, such as patient questionnaires, are prone to recall bias and may overestimate adherence. Studies comparing self-reported data with sensor-based monitoring consistently reveal lower actual adherence rates [8,9]. Post-operatively, allowing patients to move within a predetermined safe arc of motion is crucial [21,22,23]. Sensors integrated with slings could help identify whether patients remain within the safe ROM, thereby preventing excessive movement. Integration with mobile devices for real-time monitoring could further support patients by providing immediate audio or visual feedback on their ROM, ensuring their rehabilitation remains safe and effective [19].

This study provided the opportunity to further validate an existing sling wear detection algorithm [2]. We were able to simplify and improve the performance of the algorithm with a state-machine that could be implemented in future online, real-time DHT [17]. The success of the approach, even when 67.1% of the wear time had a movement standard deviation below the non-wear threshold of 13 m*g*, is especially encouraging [12]. Including walking, sitting, and standing periods in future observational validation studies is recommended to further validate the classification of arm posture during sling wear. This would help confirm that high-movement standard deviations in neutral and external rotation postures are due to walking.

The addition of a wrist-worn sensor did not improve the models for sling wear but would support a more detailed kinematic assessment of arm posture [14]. In addition, a wrist-worn sensor would provide lower arm posture data during sling non-wear and more general lifestyle information, i.e., on quality of life and sleep behaviours [24]. The data from the wrist sensor alone in the office environment demonstrated a visual representation of the restricted movement characteristic of sling wear. As a next stage, longer-term wear data are needed to develop a quantitative approach to predicting sling adherence from just wrist sensor data. While this study utilised 100 Hz, data collected at a lower frequency would suffice and would enable data collection over prolonged periods [25].

Our study had several limitations to be addressed in future work. It only included a small number of healthy volunteers monitored for less than 24 h and only in an office environment. Notably, our analysis revealed large between-participant differences of arm posture that may not have been indicative of the target population, who most likely had some movement impairment. Future studies with more participants selected from the target population over longer periods will provide more accurate descriptive summaries of typical arm posture distributions. However, the fundamental biomechanical constraints will be unchanged and therefore the basic models will remain valid. While a secure sensor pocket was stitched into the sling, it is possible that there was some variation in placement of the sensor within the sling. To minimise this, fixing of the sensor within the sling must be standardised and controlled to minimise any variance introduced.

## 5. Conclusions

While this study builds on previous work to assess sling wear adherence, to our knowledge, it is the first to assess arm postures while wearing a sling using triaxial accelerometers.

Our preferred model for assessing arm postures with a sling-embedded accelerometer is simple to implement. It was a good predictor of upper arm angle and arm posture type during sling wear (R^2^ = 0.79, *p* < 0.001 and Cohen’s kappa = 0.886, respectively). We were also able to reproduce and optimise the previously described sling wear adherence algorithm to achieve an accuracy of 99%. With this advanced methodology, we are now able to assess both components of adherence to sling wear (sling wear duration and posture of the arm when wearing the sling) continuously and automatically during everyday living.

This digital health technology has the potential to deliver evidence to inform ongoing consensus seeking in post-operative rehabilitation following shoulder surgery. It will allow the associations between patient outcomes and sling wear to be explored with objective data. The technology provides the basis for future studies aiming to define sling wear recommendations for improved patient recovery times and better functional capacity outcomes. The accurate assessment of both the sling wear duration and degree of immobilisation will support recommendations that can include considerations of prehabilitation, repair type, and rehabilitation programme adherence.

## Figures and Tables

**Figure 1 sensors-25-00166-f001:**
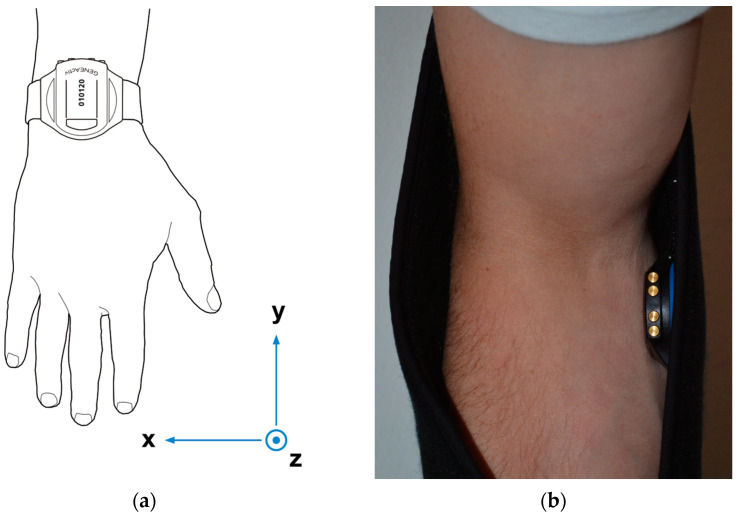
(**a**) Position of the GENEActiv fitted to the inner side of the sling next to the participant’s arm; (**b**) orientation of axes.

**Figure 2 sensors-25-00166-f002:**
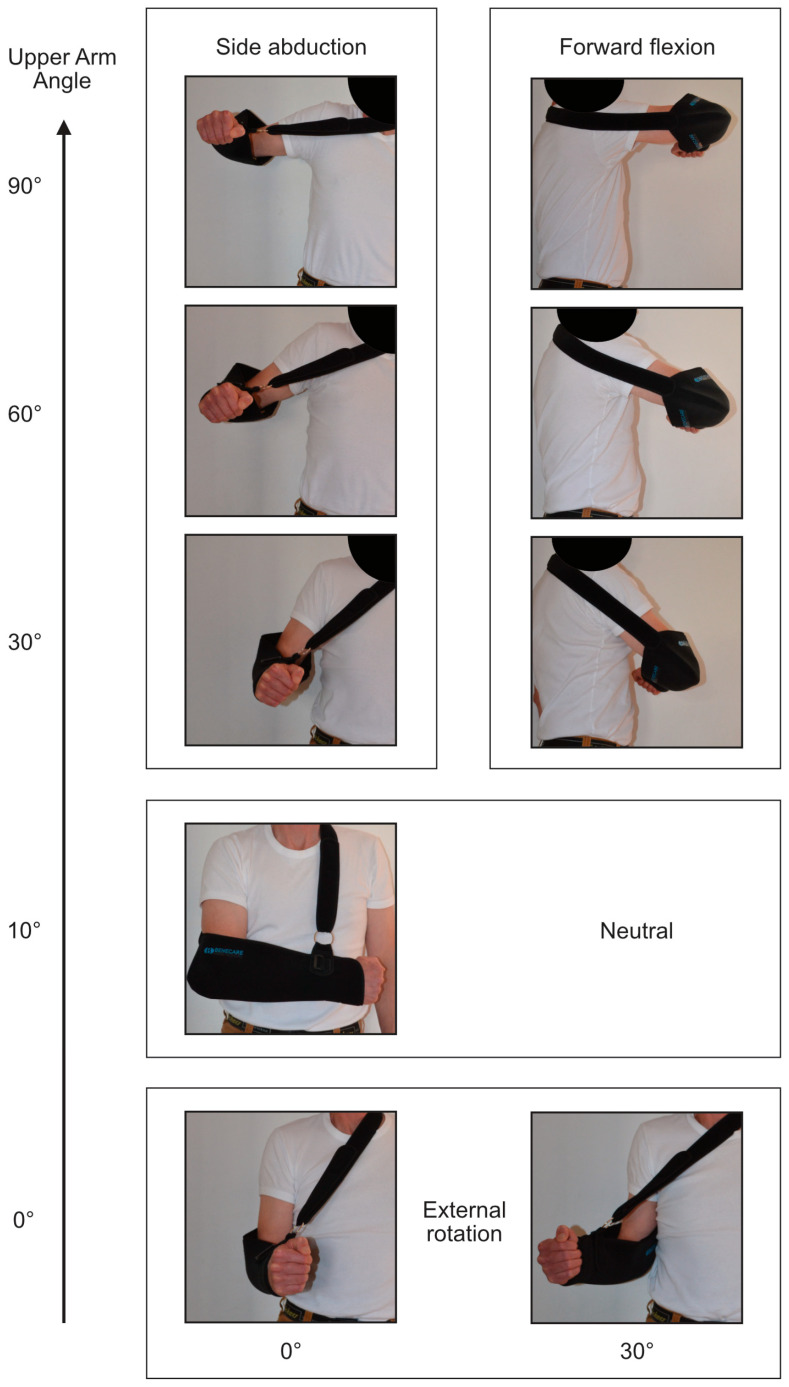
Pre-specified arm postures during sling wear. Upper arm angles: neutral (10°), forward flexion (30°, 60°, 90°), side abduction (30°, 60°, 90°); external rotation angles (0°, 30°).

**Figure 3 sensors-25-00166-f003:**
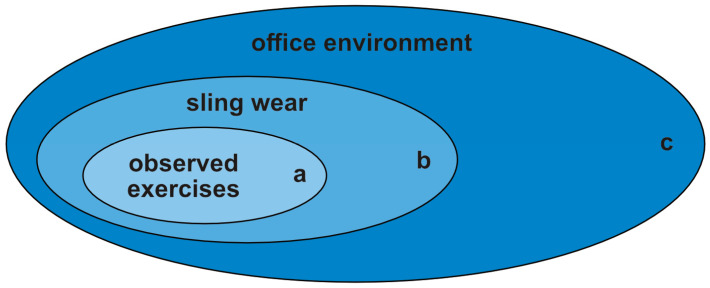
The three nested datasets generated for each participant: (**a**) sling wear during pre-specified arm postures during sling wear with recorded upper arm angle; (**b**) sling wear in free-living ‘office environment’; (**c**) sling wear and non-wear in free-living ‘office environment’.

**Figure 4 sensors-25-00166-f004:**
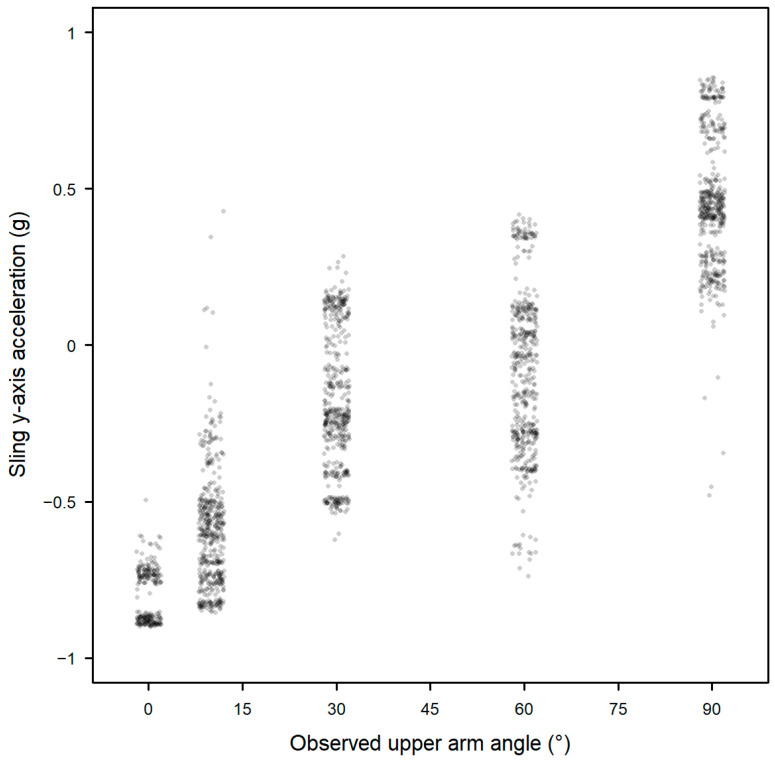
Association between upper arm angle (displayed with jitter) and acceleration in the y axis of the sling sensor (Model 1).

**Figure 5 sensors-25-00166-f005:**
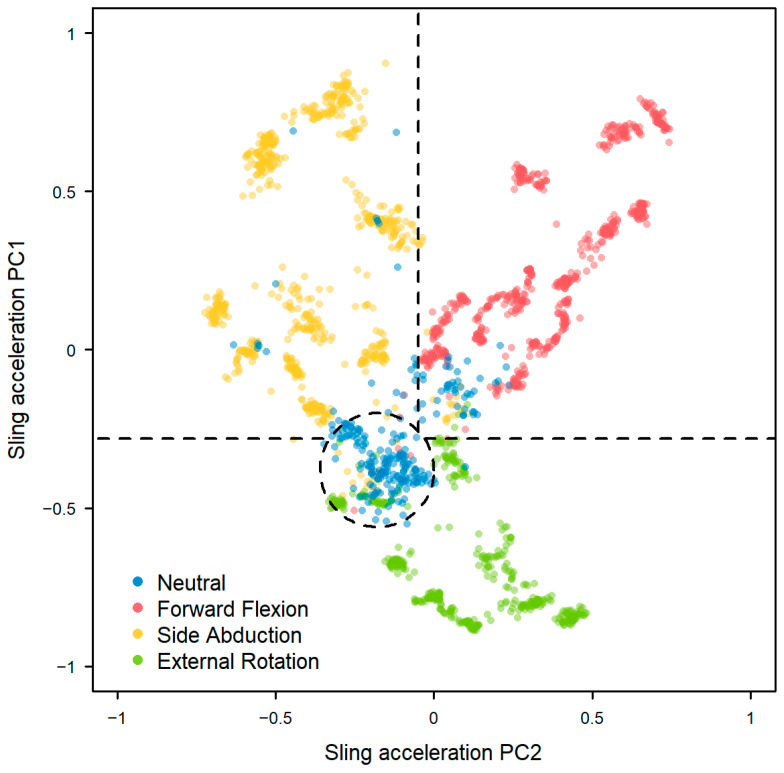
Discrimination between arm posture types using principal component analysis on sling sensor data with the dashed lines showing the boundaries of the simple classification schema.

**Figure 6 sensors-25-00166-f006:**
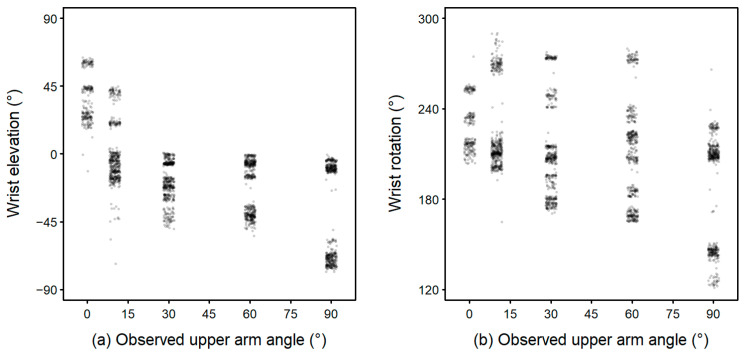
Association of upper arm angle with wrist elevation (**a**) and wrist rotation (**b**) (Model 5a).

**Figure 7 sensors-25-00166-f007:**
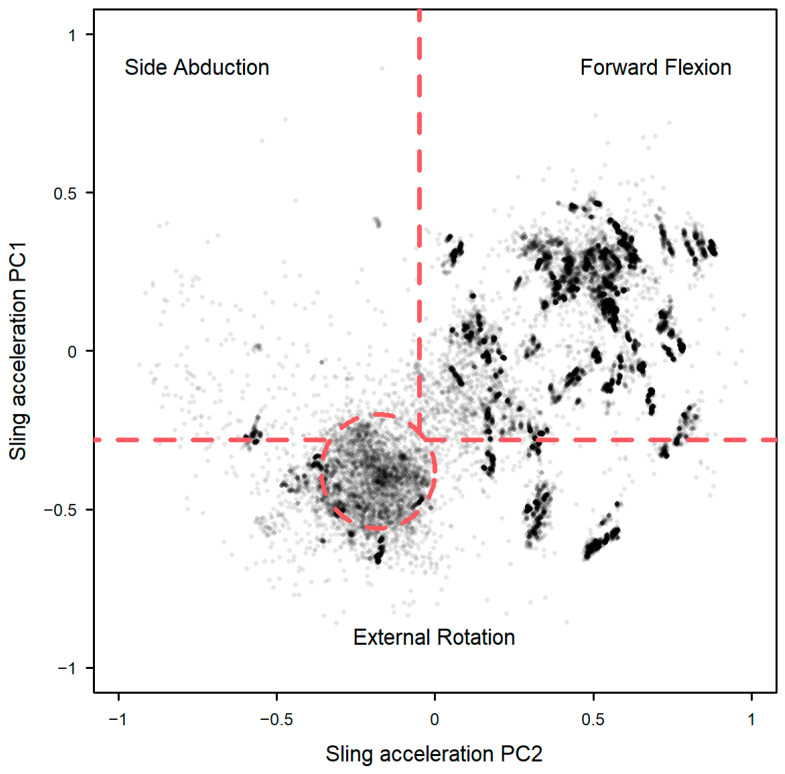
Arm posture during sling wear by second for all participants combined, with type predicted by cluster analysis on PC1 and PC2 from sling sensor data (Model 7) with the dashed lines showing the boundaries of the simple classification schema.

**Figure 8 sensors-25-00166-f008:**
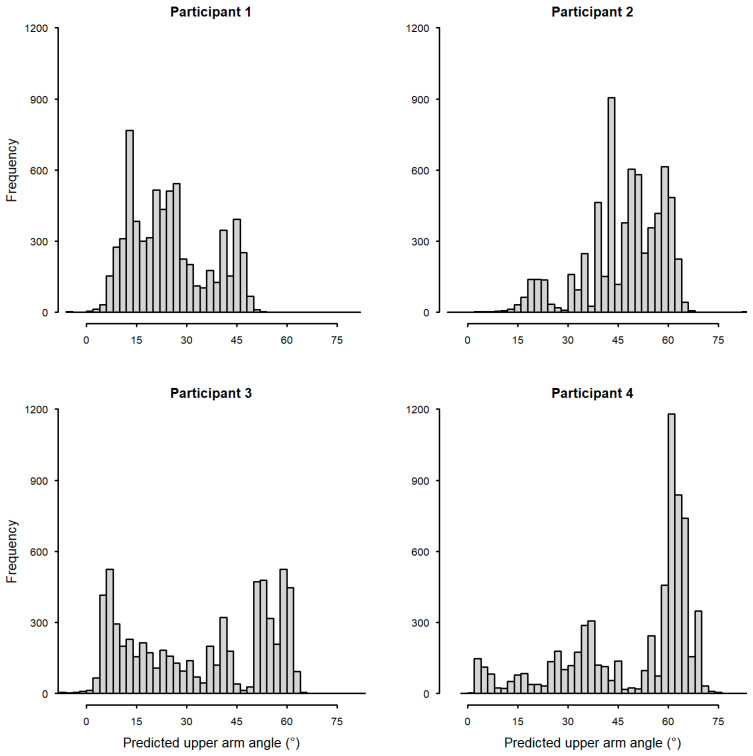
Frequency of upper arm angle predicted from sling sensor by participant.

**Figure 9 sensors-25-00166-f009:**
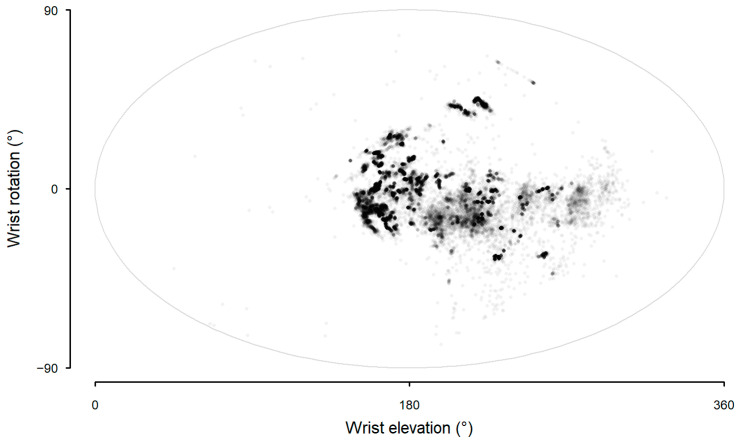
Aitoff projection of wrist elevation and wrist rotation from the wrist sensor data during sling wear in a free-living office environment.

**Figure 10 sensors-25-00166-f010:**
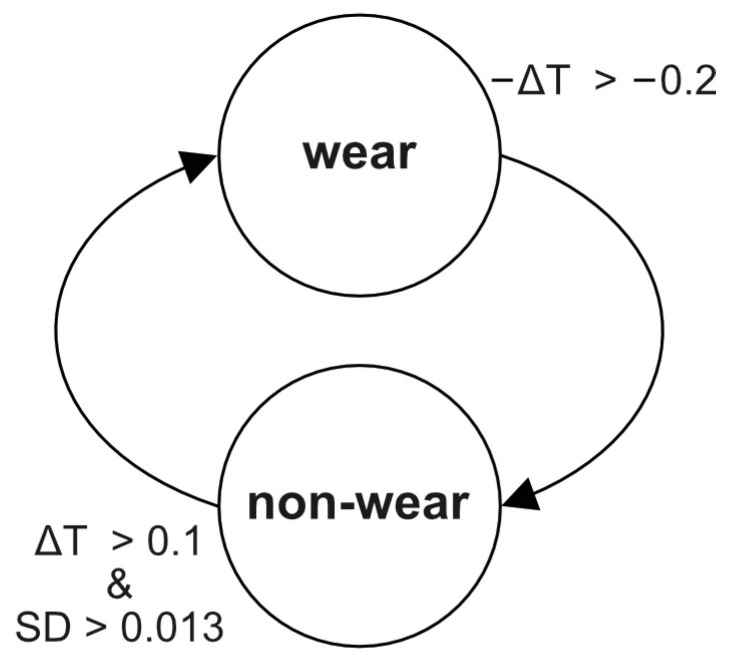
Optimised sling non-wear state machine algorithm applied to consecutive minutes. ΔT = temperature difference (°C); SD = standard deviation of movement (*g*).

**Figure 11 sensors-25-00166-f011:**
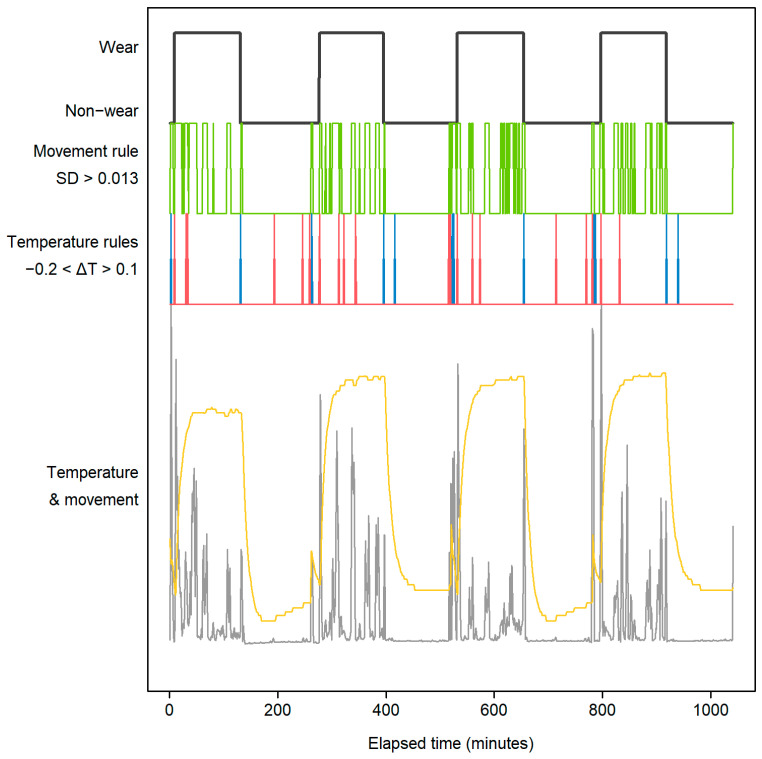
Identification of wear and non-wear for each of the 4 participants in sequence.

**Figure 12 sensors-25-00166-f012:**
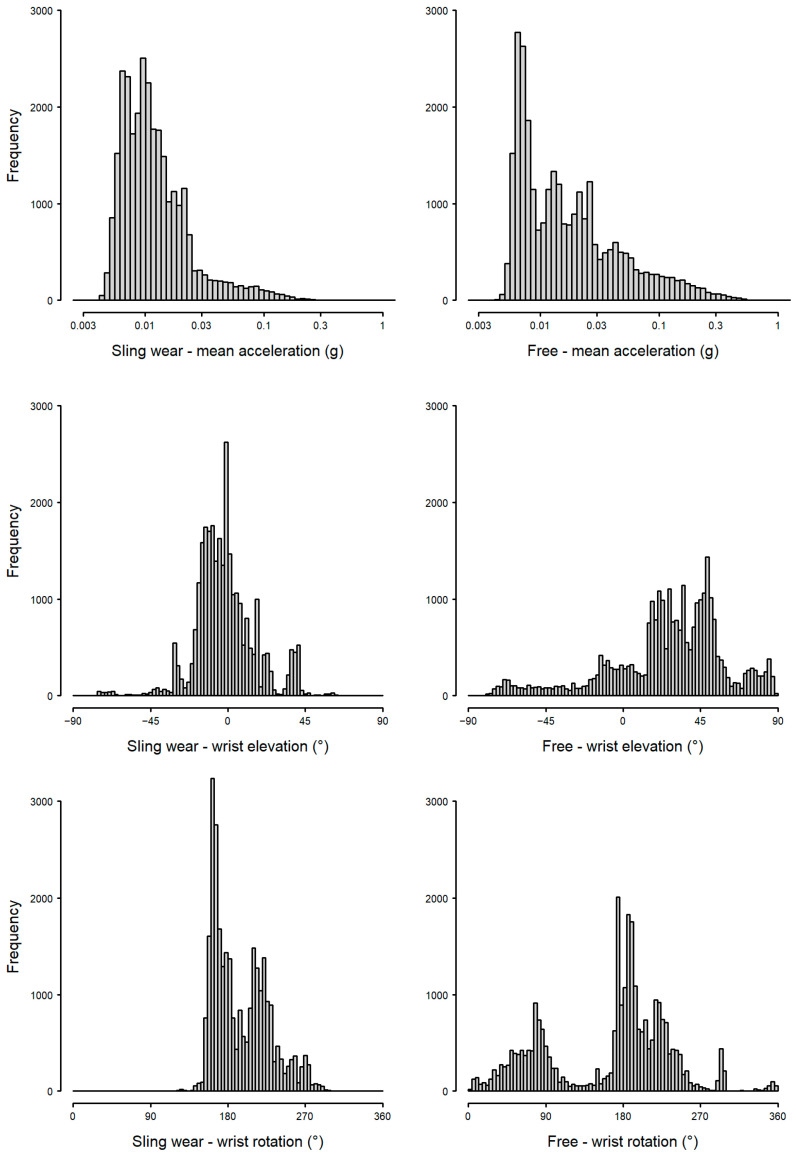
Mean wrist acceleration (**top**), wrist elevation (**middle**), and rotation (**bottom**) during sling wear (**left**) and free movement (**right**).

**Figure 13 sensors-25-00166-f013:**
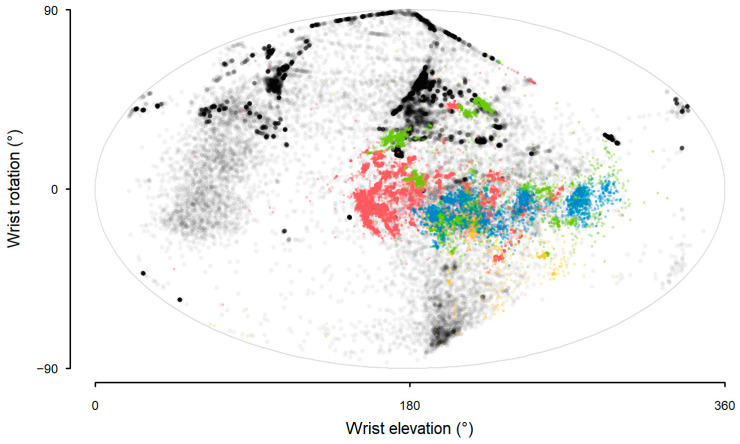
Aitoff projection of wrist elevation and wrist rotation from the wrist sensor data during sling non-wear (black with transparency) and sling wear categorised by predicted arm posture during sling wear: neutral (blue), forward flexion (red), side abduction (yellow), and external rotation (green).

**Table 1 sensors-25-00166-t001:** Principal component analysis for prediction of upper arm elevation for mean acceleration in x, y, and z axes.

	PC 1	PC 2	PC 3
Relative importance of each component			
Per component	57.7%	32.7%	9.6%
Cumulative	57.7%	90.4%	100%
Loadings of mean acceleration in x, y, and z axes on each component			
X axis	0.92	0.37	0.15
Y axis	−0.39	0.82	0.42
Z axis		0.45	−0.90
Variance explained in upper arm angle			
Per component	79.2% *	0.03%	
Cumulative	79.2% *	79.2% *	

* *p* < 0.001.

**Table 2 sensors-25-00166-t002:** Relative time spent in different predicted sling postures with mean standard deviation of acceleration in each posture.

	Relative Time (%)	Mean SD (m*g*)
Neutral	12.3	39.9
Forward flexion	67.1	12.1
External rotation	18.5	20.1
Side abduction	2.0	47.9

**Table 3 sensors-25-00166-t003:** Predictive performance of original Baraket et al. [2] sling adherence algorithm and its optimised form.

	Original Algorithm	Optimised Algorithm
Accuracy	84.6%	99.3%
Sensitivity	71.6%	98.8%
Specificity	100%	100%
F-score	0.83	0.99

## Data Availability

The data that support the findings of this study are not openly available due to containing information that could compromise research participant privacy/consent. Requests for participant-level quantitative data and statistical codes should be made to the corresponding author.

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
