# Peer review of "Digital Health Technologies for Optimising Treatment and Rehabilitation Following Surgery: Device-Based Measurement of Sling Posture and Adherence"

_sensors, 2024, doi:10.3390/s25010166_

Round 1
Reviewer 1 Report
Comments and Suggestions for Authors
Lines 133-134 should be moved to the first paragraph of Section 2 (Materials and Methods) for better flow and clarity.
Why were only 4 participants included in the study? Please provide a justification for this sample size.
What was the rationale for selecting the specific placement of the sling sensor on the participants? Since the sling sensor was in contact with their arms, did any participants report discomfort after prolonged use?
Were the 9 arm postures randomized among the participants, or did they perform the postures in a specific sequence?
What criteria were used to define and select a healthy volunteer?
What was the process for recruiting the volunteers? Additionally, were the participants compensated for their involvement?
Images should be positioned close to their first citation in the text for better readability. For example, Figure 1 is placed far from where it is first mentioned.
Device Specifications:
Does the GENEActiv device store accelerometer data locally, or does it send the data to the cloud?
If it stores it locally, what is its storage capacity, and how many days of data can it hold at once?
What is the justification for using 1-second epoch summaries in the analysis?
Why were the specific four features derived from the epoch summaries chosen?
Was the sensor capturing body temperature or environmental temperature? What unit of measurement was used?
How did participants ensure their arms were positioned accurately at the specified angles (30°, 60°, 90°), given that the study was conducted in an office environment without researcher supervision? It would be helpful to clarify if participants were trained beforehand.
On line 198, provide additional details about the Baraket et al. algorithm and the specific optimizations applied to it.
Specify the platform used for developing and training the models.
At what point in the analysis were the temperature dataset and the standard deviation data features utilized?
What is the interpretation of the R² increase from 0.75 to 0.83? What does this improvement indicate in the context of the study?
In Section 3 (Results), explicitly state whether the findings support the respective hypotheses.
In the methodology section, provide details on how the temperature and movement rules were determined.
Line 274: Correct the typo.
Would the results differ significantly if the participant size was substantially increased? Please address this possibility.
Reviewer 2 Report
Comments and Suggestions for Authors
I have received the manuscript entitled " Digital health technologies for optimising treatment and rehabilitation following surgery: device-based measurement of sling posture and adherence " for review and have found it to be very interesting and very well-written.
I believe that the authors have clearly presented the general aim, hypotheses, scope, methodology, experimental settings, and results included in the manuscript. Therefore, the current state of the manuscripts is suitable for publication after some minor corrections.
I made very minor linguistic corrections, and some specific data presentation suggestions. All of these are highlighted "in YELLOW colour" inside the pdf file. By double-clicking on any highlighted text, the authors will find “inside a balloon window” a correction or a suggestion or a concern for clarification.
Exact comments / queries / suggestions to enhance the work are highlighted inside the pdf file. Most of these comments are related to results presentation and clarifications of some experimental settings.

Round 2
Reviewer 1 Report
Comments and Suggestions for Authors
N/A